# Immediate Functional Loading with Full-Arch Fixed Implant-Retained Rehabilitation in Periodontal Patients: Clinical Study

**DOI:** 10.3390/ijerph192013162

**Published:** 2022-10-13

**Authors:** Eugenio Velasco-Ortega, Joao Luis Cracel-Lopes, Nuno Matos-Garrido, Alvaro Jiménez-Guerra, Ivan Ortiz-Garcia, Jesús Moreno-Muñoz, Enrique Núñez-Márquez, José Luis Rondón-Romero, José López-López, Loreto Monsalve-Guil

**Affiliations:** 1Department of Comprehensive Dentistry for Adults and Gerodontology, Faculty of Dentistry, University of Seville, 41018 Seville, Spain; 2Department of Odontoestomatology (Dentistry), Service of the Medical-Surgical Area of Dentistry Hospital, University of Barcelona, 08907 L’Hospitalet de Llobregat, Spain

**Keywords:** immediate loading, immediate functional loading, dental implants, fixed rehabilitation, periodontal patients

## Abstract

(1) Background. The immediate functional loading of implants is a clinical procedure used for treating periodontal edentulous patients. This clinical study aimed to evaluate the clinical outcomes of the immediate functional loading of implants with fully fixed rehabilitations in compromised periodontal patients. (2) Methods. Three hundred and five implants IPX screw implants were placed in 27 periodontal patients using an immediate functional loading protocol with fixed rehabilitations. All patients had a previous history of periodontitis, four patients (14.8%) were smokers and seven patients (25.9%) suffered from chronic medical conditions. (3) Results. Implant and prosthetic clinical findings were evaluated during a mean period of 41.3 ± 19.6 months. No implants were lost during the clinical follow-up. The cumulative survival rate for all implants was 100%. Regarding the prostheses designed, a total of 54 fixed prostheses were placed in the 27 patients immediately after the surgery. Forty-four hybrid fixed prostheses (81.5%) and 10 fixed rehabilitations (18.5%) were placed in the patients. The mean marginal bone loss was 1.51 ± 1.16 mm, ranging from 0 to 3.5 mm during the follow-up evaluation. Thirty-one implants (10.2%) in 10 patients (37%) were associated with peri-implantitis. Five patients (18.5%) showed some kind of technical complications (loss/fracture of the prosthetic screw, acrylic resin fracture, ceramic chipping). (4) Conclusions. The clinical outcomes of this study demonstrate that fixed rehabilitation by immediate functional loading of implants is considered a predictable procedure.

## 1. Introduction

Edentulism is the final marker of the disease burden for oral health and remains a major oral disease worldwide. Many million people globally have been affected by edentulism and severe tooth loss. Prolonged total or partial edentulism is associated with the progressing resorption of alveolar processes. Additionally, many dental problems are related to edentulism. The scientific evidence suggests a relationship between TMDs and a wear dental occlusion. Patients with TMD symptoms often need comprehensive prosthetic treatment, including partial edentulism, esthetic deficiencies, and functional problems. Variable options are available for partially and edentulous patients, starting from restoring oral functions using removable dentures or utilizing dental implants for fixed prostheses [1,2].

Implant dentistry is a long-term good solution for the prosthetic rehabilitation of partially and edentulous patients, with a high rate of survival and success. Implant-supported overdentures have been demonstrated to be an effective treatment option for restoring patients with completely edentulous arches. A two-implant overdenture has been recommended as the first choice of treatment for patients with an edentulous mandible, with a high level of patient satisfaction and quality of life [3]. However, in fully edentulous patients, rehabilitation with fixed implant-supported prostheses can be an effective and better treatment to restore aesthetics and chewing function, resulting in the significant improvement in their quality of life [4,5]. The importance of diagnostics and treatment planning for an adequate fixed rehabilitation requires an implant restoration design. Edentulism may be treated successfully using a treatment approach involving four, six, or more implants [2,5].

Full-arch fixed dental prostheses present high survival and success rates with all loading protocols (conventional, early, and immediate) [5,6,7,8,9]. In last years, an important number of clinical studies and systematic reviews demonstrated that the early and immediate functional loading of dental implants can be as effective as those treated with conventional loading protocols [5,6,7,8,9]. Several systematic reviews evaluated loading protocols in edentulous patients with fixed implant-supported prostheses and showed a similar rate of implant survival, failure, and complications, regardless of the loading protocol when treating the maxillary and mandibular arch [5,9].

The immediate functional loading protocol is characterized by the delivery of the prosthesis within 1 week of implant placement with a minimum torque of 35 N cm [10]. The clinical success of immediate functional loading is highly dependent on several factors: patient selection, bone quality and quantity, implant number and design, implant primary stability, and occlusal loading. Implant primary stability is undoubtedly the most important factor [11]. Several advantages are related to the immediate loading of dental implants as the reduction of time, the improvement of esthetic and occlusal function, the exclusion of temporary removable prostheses, the prevention of second surgeries, and the preservation of residual alveolar ridges [12,13]. The rehabilitation of full edentulous patients by using complete-arch fixed prosthesis with several implants has been further developed applying immediate-function protocols with the connection of the prosthesis on the same day of the surgery. The rehabilitation of full edentulous jaws through the immediate-function loading of a fixed prosthesis supported by implants is considered a predictable procedure [14,15,16,17].

Implant treatment in compromised periodontal patients has been suggested to have a different outcome when compared with patients without a history of periodontitis. Lower survival rates, higher marginal bone loss, and the prevalence of peri-implantitis are associated with the use of implants in patients with periodontitis [18,19]. A recent retrospective study evaluated the longevity of teeth and implants over a long-term period (≥10 years) in periodontal patients [20]. Fifty-eight periodontal patients who had received periodontal therapy and maintenance were included. Periodontal clinical and radiographic parameters were assessed at six tooth or implant sites. The implant failure rate was 10.08%, and the implant failure rate due to biological reasons was 9.8%. The incidence of implant failures in patients with vs. without recurrent periodontal disease was 83.3% vs. 16.7%. The clinical outcomes showed that chronic periodontitis patients are successful in keeping the majority of periodontally compromised teeth, but a higher tendency for implant loss than tooth loss was found [20]. A long-term clinical study conducted over 20 years, concerning dental implants in patients with and without a history of periodontitis, showed that implants, placed after periodontal treatment and supportive periodontal care, yield favorable long-term results. However, patients with a history of periodontitis and non-compliance with supportive periodontal care are at higher risk of biological complications and implant loss [21].

The immediate functional loading of an implant-supported fixed complete denture is a suitable treatment option for edentulous patients with a history of periodontitis with high survival implant rates [22]. Several clinical studies suggest that the immediate functional loading of implants placed in immediate function in post-extraction sockets in periodontally compromised patients may be provided by a surgical and prosthetic protocol [23,24]. Good results have been documented with immediate implant function after extractions in periodontal sites with the same level of success as in non-compromised sites [23]. A clinical retrospective study included the participation of edentulous patients with a minimum follow-up period of 6 years. All the implants survived successfully, demonstrating that the immediate functional loading of implants proves the long-term stability of the prosthodontic rehabilitation of edentulous jaws with a higher success rate in patients with a background of periodontitis [24]. The maintenance is very important for the long-term success of the immediate functional loading of implants in edentulous patients with a history of periodontitis. Clinicians should pay more attention to the regular maintenance of compromised periodontal patients to reduce biological and mechanical complications [22,23,24].

This clinical study aimed to evaluate the clinical outcomes of immediate functional loading of implants with fully fixed rehabilitations in periodontally compromised patients.

## 2. Materials and Methods

### 2.1. Sample Description

This study included patients with a history of periodontal disease, based on the classification of Caton et al. [25] and Pakdeesettakul et al. [26], with total or partial edentulism that requires the extraction of all remaining teeth and treatment with immediate dental implants. All surgeries and prosthetic procedures were developed in the School of Dentistry of Seville University, Spain, from January 2019 to December 2020. The study was conducted according to the principles outlined in the Declaration of Helsinki [27] on clinical research involving humans. All patients signed a double informed written consent for implant placement and were part of the clinical study. The ethical committee of the University of Seville approved the study.

#### 2.1.1. Demographic Distribution

Twenty-seven patients were included in the study; 13 males and 14 females of ages ranging from 47 to 91 years old, with a mean age of 64.7 ± 10.6 years old.

#### 2.1.2. Inclusion and Exclusion Criteria

The inclusion criteria were: adult patients, good systemic health status (ASA I or II) or controlled systemic diseases, and no need for bone regeneration techniques prior to implant placement surgery.

The exclusion criteria were the presence of uncontrolled chronic systemic disease (diabetes, cardiovascular disease), smoking more than ten cigarettes per day, coagulation disorders, alcohol or drug abuse, and the use of any medication or health alteration that contraindicates implant treatment.

### 2.2. Diagnosis and Treatment Plan

Treatment planning included diagnostic casts to evaluate intermaxillary relations, clinical photographs, and panoramic radiographs (Figure 1). Most of the patients were evaluated with computerized beam cone tomography when required (Figure 1).

### 2.3. Surgery Protocol

All patients received prophylactic antibiotic therapy one hour before surgery (500 mg amoxicillin and 125 mg clavulanic acid) and continued to take the antibiotics plus 600 mg ibuprofen postoperatively; one capsule every eight hours for seven days. Based on the clinical performance criteria of our working group [4] and endorsed in part by the recent literature [28], the use of chlorhexidine mouthwash was recommended twice daily for one month. All patients were treated under local anesthesia with articaine and adrenaline.

Before the implant surgery, all remaining periodontal teeth were extracted. A mucosal flap approach was made, and the implants were inserted in the selected place following a prosthodontic-guided plan. The drilling protocol was the one recommended by the manufacturer (Galimplant^®^, Sarria, Spain), and the minimum insertion torque was 35 N cm (Figure 2). All implants were inserted into a healed bone and immediately after extractions using a one-stage surgical technique. Bone bovine graft Cerabone^®^ (Bottiss Biomaterials GmbH (Zossen, Germany) was applied when it was necessary. A collagen membrane (Bio-Gide Geistlisch Pharma AG, Wolhusen, Switzerland) covered these augmented areas

After implant placement, prosthetic abutments were immediately placed and functional loading was completed when the insertion torque achieved at least 35 N cm. Implant-supported fixed prostheses were placed immediately after surgery (Figure 3).

### 2.4. Follow-Up

After a control visit at 7 and 21 days for a postsurgical evaluation, follow-up visits were scheduled at 3 and 6 months after prosthesis placement and every year during a mean period of 41.3 ± 19.6 months (ranging between 15 and 91 months). The success criteria were established as implant stability and the absence of radiolucency around the implant, mucosal suppuration, or pain. Marginal bone loss was determined by an intraoral digital radiograph taken perpendicular to the long axis of the implant.

### 2.5. Implant Characteristics

IPX screw implants (Galimplant^®^, Sarria, Spain) were used for all patients. The implant surface has been treated with sandblasting and etching acid (SLA) to increase the roughness inducing a microtopography. This surface, treated with a defined treatment method, resulted in a rougher surface, with Ra between 1.5–2.0 µm. The implant was a tissue-level, commercially pure (CP) titanium grade IV implant characterized for an internal hexagon connection.

### 2.6. Statistical Evaluation

The software SPSS 18.0 (SPSS Inc., Chicago, IL, USA) was used for data evaluation. Descriptive statistics were used to describe the results as mean ± standard deviation. The chi-squared test and a two-way ANOVA with a U-Mann–Whitney test were used for statistical analysis, establishing the level of significance at *p* < 0.05.

## 3. Results

Three hundred and five implants were placed in 27 bimaxillary edentulous patients, 13 males and 14 females, with an average of 11.3 implants/patient. No significant statistical differences were found related to sex and age (chi-square test, *p* = 0.33169). All patients had a previous history of periodontitis, four patients (14.8%) were smokers and seven patients (25.9%) suffer from chronic medical conditions (Table 1). The mean age of the women was 63.7 ± 10.2 years and that of the men was 65.6 ± 11.4 years. These differences were not significant according to the analysis of variance (ANOVA; *p* = 0.6513).

Four patients were smokers (less than 10 cigarettes/day) and had 45 implants inserted (14.7%); the distribution by age, sex, and medical history is presented in Table 2. Patients with a well-controlled medical history (hypertension, heart failure, diabetes) represented seven patients (25.9%) with 82 implants (26.9%) inserted (Table 3).

Of the 305 implants placed, 200 (65.5%) had a diameter of 3.5 mm and 105 (34.5%) had a diameter of 4 mm. Five implants (1.6%) were 8 mm in length, twenty-five (8.2%) were 10 mm, one hundred and ninety-five (63.9%) were 12 mm, and eighty (26.2%) were 14 mm. One hundred seventy implants (55.7%) were inserted in the maxilla, and one hundred and thirty-five implants (44.3%) were placed in the mandible. Regarding the bone filling material, it was used in 26 of the 27 patients, and of these, 13 patients were under 65 years of age and 14 were not, with a *p* = 0.32611. In three patients (11.1%) a maxillary sinus lift was performed simultaneously with the placement of the implants; not showing significant differences in terms of sex and age (Table 4). No implants were lost during the clinical follow-up. The cumulative survival rate (CSR) for all implants was 100%.

Regarding the prostheses designed, a total of 54 fixed prostheses were placed in the 27 patients immediately after the teeth extractions. Forty-four hybrid fixed prostheses (81.5%) and ten fixed rehabilitations (18.5%) were placed in the patients. In the upper jaw, 19 total hybrid prostheses (70.4%) and 8 fixed restorations (29.6%) were performed. In the mandible, 25 total hybrid prostheses (92.6%) and 2 fixed restorations (7.4%) were performed. There were no significant differences in age but there were regarding sex, with a *p* = 0.0165 (Table 5).

The mean marginal bone loss was 1.51 ± 1.16 mm, ranging from 0 to 3.5 mm during the time interval from the implant insertion to the 3.5-year follow-up evaluation. In patients with a chronic medical condition, this marginal bone loss was 2.14 ± 1.02 mm, while in patients without systemic disease it was 1.30 ± 1.15 mm. These differences show statistical significance (U-Mann–Whitney test; *p* = 0.0486). Regarding smoking habits, the marginal bone loss was 2 ± 1.58 mm for smoking patients and 1.43 ± 1.10 mm for non-smoking patients, with no statistical differences (U-Mann–Whitney test; *p* = 0.4105). Neither age nor gender showed significant differences (Table 6).

During the follow-up period, 31 implants (10.2%) in 10 patients (37%) were associated with peri-implantitis [25]. The peri-implantitis was more frequent, showing statistically significant differences in those patients with a chronic medical condition (87.5%) (Chi-square test, *p* = 0.0438). The peri-implantitis was also more frequent in smoking patients (100%) without statistically significant differences (Chi-square test, *p* = 0.14766, and *p* = 0.58596, respectively) (Table 7). Five patients (18.5%) showed some kind of technical complications (loss/fracture of the prosthetic screw, acrylic resin fracture, ceramic chipping).

## 4. Discussion

This study evaluated the clinical results of the treatment of patients with periodontal disease, who were fully edentulous during implant placement surgery and full-arch rehabilitation with the immediate functional loading of implants placed post-extraction or in a previously healed bone. The full rehabilitation of edentulous jaws is an important challenge because optimal implant planning is strongly based on the radiographic data of bone availability for an accurate approach to prosthetics [22]. This retrospective clinical study assessed implant survival rates on the implant and patient-related level of implant-supported immediately loaded fixed full-arch rehabilitation in compromised periodontal patients with a follow-up of up to 3.5 years. The CSR of implants being placed in this study yielded 100% and is comparable to several long-term results of rehabilitation procedures with immediately loaded treatment approaches of the edentulous jaws with CRS between 95–100% [7,14,20,22].

The increased popularity of immediate functional loading among dentists can be explained by the reduction in treatment time. The patient and the professional have a growing interest in shortening the time between implant placement and rehabilitation with a functional and esthetic prosthesis that provides faster comfort and social well-being. However, time management should not be decisive in choosing this treatment clinical protocol. Only a comprehensive diagnosis and treatment planning of the patient must be established according to surgical skills, prosthetic quality, and long-term maintenance [11,12]. According to recent clinical studies, implants placed with an immediate functional loading with fixed full-arch prostheses reported a very high success rate after several years of follow-up, both in the fresh sockets and healed sites [14,19,20,24,29,30].

Another attempt to ease implant therapy for the patient involved efforts to reduce the time between tooth extraction and implant placement. Implant placement immediately following a tooth extraction is a frequent and predictable clinical procedure and is considered as placing implants into healed sites [23,24]. In the present study, before the implant surgery, remaining periodontal teeth were extracted. All implants were inserted simultaneously into a healed bone and immediately into fresh sockets. Covani et al. [31] evaluated the outcome of treatment in the rehabilitation of edentulous jaws with immediate loaded full-arch screw-retained prostheses after up to 4 years of function. A total of 19 patients with completely edentulous maxillae and/or mandibles or presenting natural teeth with a poor or hopeless prognosis received six implants each in the mandible and/or eight in the upper jaw. All patients received a full-arch prosthetic reconstruction. A total of 164 implants were inserted, 119 implants were placed immediately after tooth extraction, and 45 implants were placed in healed sites. Overall, eight implants failed, leading to a 4-year cumulative survival rate of 95.1%. Polizzi et al. [32] evaluated the survival rate of patients with compromised dentition treated with immediately fixed restorations on maxillary implants inserted in fresh extraction and healed sites. Twenty-seven patients were treated with flapless surgery. Immediate full-arch (*n* = 19) or partial (*n* = 10) restorations were delivered. Patients were followed for up to 5 years. One hundred sixty implants were assessed. Four implants in two patients failed and were removed (overall CSR 97.33%) and two were replaced. All final prostheses were functionally stable, demonstrating a good outcome concerning implant and prosthesis survival [32].

Today, a treatment protocol for patients with advanced periodontal disease includes the extraction of periodontally involved teeth and immediate implant placement, followed by restoration with fixed dental prostheses—this process shows a high survival rate [33]. However, there is an important controversy on the general use of implants in patients with periodontitis. Several studies report a higher risk for marginal bone loss, peri-implantitis, and implant failure in these patients [34,35]. In the present study, all patients had a previous history of periodontitis and four patients (14.8%) were smokers. Some studies have reported the positive clinical outcomes of fully edentulous patients treated with the immediate functional loading of immediately placed implants in periodontally compromised patients [23,24,36,37,38]. A study on full-arch immediate implant and restorations in patients with advanced generalized aggressive periodontitis was designed to evaluate the clinical outcomes after an average of 5 years [38]. Seventeen patients received immediate post-extraction implants and rehabilitation. Eighty implants were inserted into 20 arches (seven maxillae and thirteen mandibles). The CSR of the implants was 98.75% (79/80). One tilted implant failed due to peri-implantitis. The CSR was 100% (20/20) for definite prostheses, while 85% (17/20) for provisional prostheses. Patients showed high satisfaction with the overall effects [38].

In the present study, a xenograft (bovine bone) was applied inside and outside the sockets covered by a resorbable collagen membrane. This surgical approach may have contributed to the high success rate of the implants. It has been suggested that for the long-term success of a surgical protocol involving the immediate placement a bone augmentation technique with the use of an osteoconductive bone substitute is necessary, as is a resorbable membrane to prevent an extensive bone remodeling of the edentulous ridge [39,40]. Alveolar ridge volume has been improved with the use of different bone substitutes (allografts, xenografts, alloplastics) demonstrating a limited vertical bone loss. Moreover, the lower resorbability of the grafting material can also be advantageous, as it minimizes the resorption of the buccal bone [39,40]. A 5-year retrospective study reported the clinical outcomes of eighty-four axial and forty-six tilted immediate implants placed in the extraction sockets of 23 patients according to a four to six implant protocol combined with ridge augmentation and immediate functional loading [40]. The CRS of the straight and tilted implants was 100% and 97.8%, and the prosthetic was 100%. After the implant placement, the sockets and the ridge were augmented with a particulate freeze-dried bone allograft, which filled the residual gaps around the implants covered with a collagen membrane [40].

In the present study, a total of 54 fixed prostheses were placed in 27 patients immediately after the implant insertion. Forty-four screw hybrid fixed prostheses (81.5%) and ten cemented fixed rehabilitations (18.5%) were placed in the patients. In our study, we did not obtain differences in the data analyzed between one type or another of prostheses, either because there is no statistical difference or because the power of the sample is low. In most similar studies the implants were immediately restored with screw-retained restorations as an acrylic provisional and finally restored with acrylic-metal hybrid prosthesis [15,24,31,41]. Screw implant restorations have the advantages of predictable retrievability and are easier to remove when maintenance, repair, or surgical interventions are required. Screw-retained implant reconstructions require a more precise, prosthetically driven placement of the implant due to the position of the access hole [42]. However, several studies reported implants immediately loaded with cemented restorations [32,40]. The advantages of a cemented restoration include the compensation of improperly inclined implants, the passivity of fit, improved esthetics, and without a screw-access hole, resulting in better control of the occlusion. An important problem is that cement excess is related to more biological complications, such as mucositis and peri-implantitis [42].

Marginal bone loss is considered an important biological and clinical parameter when evaluating the success of implant-fixed rehabilitations. It is normal for vertical marginal bone loss around implants to reach a maximum of 1 mm to 1.5 mm during the first year of functional loading [43,44]. In the present study, the mean marginal bone loss was 1.51 ± 1.16 mm after a 3.5-year follow-up. This marginal bone loss was higher than the results described in other clinical studies of immediate functional loading in fully edentulous patients with a full arch rehabilitation with a 6–9 years follow-up [24,39,40]. Smoking habits are an important risk factor for marginal bone loss in patients treated with the immediate functional loading of implants placed in periodontally compromised sites [8,23,41]. A negative influence of implant placement in smoking patients (2 ± 1.58 mm) on marginal bone loss was also found in the present study, when compared with the non-smoking patients (1.43 ± 1.10 mm). The favorable design of fixed restorations may contribute to the better maintenance of peri-implant tissues by full access to control plaque around the abutment/crown interface at a minimum of twice a year [23,24,40].

Most periodontally compromised patients treated with immediate functional loading of implants with full-arch rehabilitations were under long-term maintenance at their dentist’s clinic [8,32]. After one year of follow-up, patients seemed rather non-compliant regarding oral hygiene measures. Many patients showed a slight amount of plaque around implant-abutment interfaces, accounting for a cumulative plaque score. This may indicate a risk for peri-implant diseases [8,32]. In the present study, patients or implants were diagnosed with peri-implantitis based on bleeding on probing in combination with marginal bone loss exceeding 2 mm. The occurrence of peri-implantitis is very high, involving 37% of the patients and 10.2% of the implants compared to 13% of the patients and 5.4% of the implants in another study that used a similar protocol and immediately loaded implants with a screwed-retained prosthesis followed for 5 years [40]. Additionally, biological complications, such as bleeding on probing were reported in 19.8% of patients in a 5-year study. Interestingly, smoking was associated with the occurrence of biological complications [41]. These clinical findings confirmed that periodontal background, poor oral hygiene, and smoking habits are important risk factors for peri-implant diseases in patients treated with immediate functional loading of implants with full-arch rehabilitation [8,18,23,32,40,41]. Finally, our results are higher than those presented in a retrospective study of more than 5000 implants [36], and some of the determining factors for other authors, regarding a higher risk of peri-implantitis, such as tobacco and poor hygiene, are also present in our study. All cases of peri-implantation could be treated with conservative measures based on the recommendations provided by Hussain et al. [37].

Both metal–acrylic resin (hybrid) and fixed full-arch metal–porcelain prostheses are possible solutions for the immediate functional loading of implants in edentulous patients. The cumulative survival rate of the prosthesis is very high until 100% [39,45]. However, mechanical–technical complications are frequent, with high percentages that increase over years of use [46,47]. Moreover, sufficient evidence suggests that prosthetic design (i.e., cantilever) may be considered that possible additional risk factors, such as parafunctional habits (bruxism) or antagonists, were not evaluated [15,41]. In the present study, five patients (18.5%) showed some kind of technical complications (loss/fracture of the prosthetic screw, acrylic resin fracture, ceramic chipping). The problems were solved based on the criteria of the group. The loss of the prosthetic screw was replaced easily, and the crown fracture was partially repaired in the clinic and in two cases it had to be sent to the prosthetic laboratory to repair the fractured ceramic. The loosening was solved with a new prosthetic screw and a torque of 25 Nw. A 5-year study of the rehabilitation of a completely edentulous mandible by using complete-arch fixed prosthesis reported mechanical complications in 27.1% of cases evaluated, demonstrating the association between bruxism and the incidence of mechanical complications because the continuous stress forces applied on the prosthesis could be related to the loosening or the fracture of any prosthetic component [41].

In terms of the limitations of the study, we can comment first of all that some patients/implants had only a short follow-up period of up to 15 months, which could influence the interpretation of the results presented. In addition, due to the design itself, there are also limitations regarding different types of prostheses, materials, and forms of retention and installation protocols in healed sites and fresh sockets; aspects that represent a heterogeneity that can negatively impact the analysis of the results. On the other hand, as a positive element, the action protocol is always the same, and the type of patient is also the same; thus, the results obtained are encouraging when referring to patients with a higher risk of failure—patients with previous periodontal pathology.

## 5. Conclusions

The immediate functional loading of implants placed in fresh sockets and healing sites can be used as an alternative treatment to the prosthodontic rehabilitation of full edentulous patients. This study indicates that the treatment of periodontal edentulous patients with full fixed rehabilitation of jaws by a clinical protocol of post-extraction implant placement and immediate functional loading appears to be a successful implant treatment. Although longer-term studies are needed, the immediate functional loading in patients with advanced periodontal disease allows for the shortening of treatment times and results in a predictable therapeutic alternative.

## Figures and Tables

**Figure 1 ijerph-19-13162-f001:**
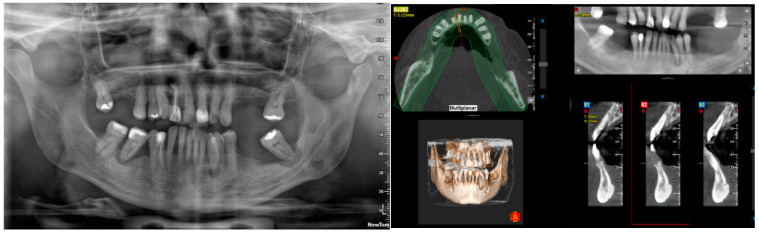
Panoramic radiograph and computerized tomography as part of the diagnosis and treatment plan.

**Figure 2 ijerph-19-13162-f002:**
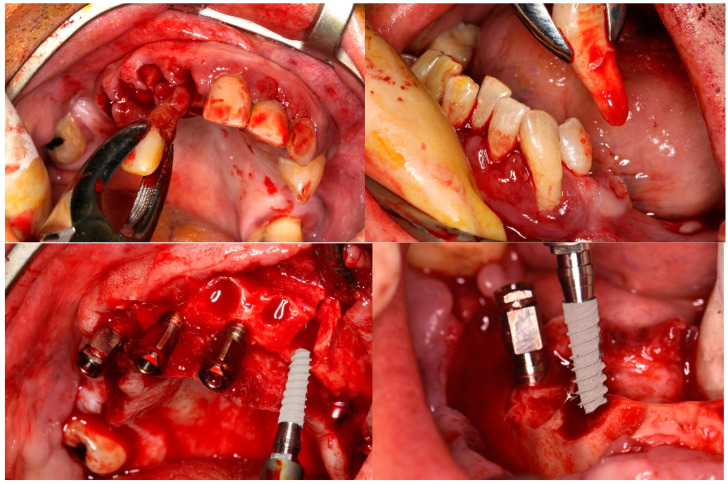
Clinical surgical protocol. Multiple tooth extraction, implant placement with mucosal flap approach, and the application of bone substitute.

**Figure 3 ijerph-19-13162-f003:**
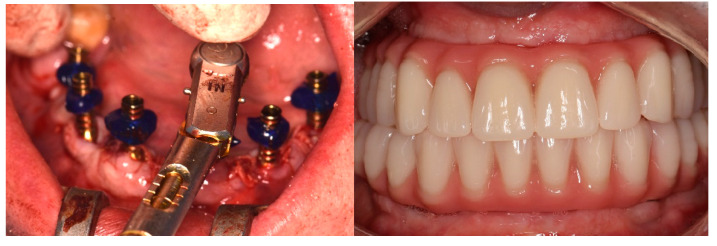
Placement of prosthetic abutments and implant-supported fixed prostheses.

**Table 1 ijerph-19-13162-t001:** Demographic and clinical variables (*n* = 26).

Variables			*p*-Value
Age	≤65 years	>65 years	0.03261 *
	13 (48.1%)	14 (51.9%)	
Sex	Men	Women	0.03261 *
	13 (48.1%)	14 (51.9%)	
Tobacco	Smokers	Nonsmokers	0.67086
	4 (14.8%)	23 (85.2%)	
Diseases systemic	+	−	0.88346
	7 (25.9%)	20 (74.1%)	
Clinical follow-up	<36 months	>36 months	0.3625
	12 (44.4%)	15 (55.6%)	

*: significant *p* value (*p* < 0.05). Test of Chi-squared.

**Table 2 ijerph-19-13162-t002:** Smoker patients (*n* = 4, 14.7%).

Variables			*p* Value
Age	≤65 years	>65 years	0.93592
	2 (15.4%)	2 (14.3%)	
Sex	Men	Women	0.00245 *
	4 (30.8%)	0 (0%)	
Diseases systemic	+	−	0.96348
	1 (14.3%)	3 (15%)	
Clinical follow-up	<36 months	>36 months	0.91231
	2 (16.6%)	2 (13.3%)	

*: significant *p* value (*p* < 0.05). Chi-squared test.

**Table 3 ijerph-19-13162-t003:** Distribution of the number of splices (*n* = 305).

Variables			*p*-Value
Age	≤65 years	>65 years	0.2816
	150 (49.2%)	155 (50.8%)	
Sex	Men	Women	0.6630
	147 (48.2%)	158 (51.8%)	
Tobacco	Smokers	Nonsmokers	0.5639
	41 (14.5%)	264 (83.5%)	
Diseases systemic	+	−	0.6239
	230 (75.4%)	31 (24.6%)	
Clinical follow-up	<36 months	>36 months	0.8656
	135 (44.3%)	170 (55.7%)	

Note: *p*-values below 0.05. Chi-squared test.

**Table 4 ijerph-19-13162-t004:** Distribution of patients with simultaneous maxillary sinus lift (*n* = 3).

Variables			*p* Value
Age	≤65 years	>65 years	0.4959
	2 (15.4%)	1 (7.1%)	
Sex	Men	Women	0.0766
	0 (0%)	3 (21.4%)	

Note: *p*-values below 0.05.

**Table 5 ijerph-19-13162-t005:** Distribution of the type of prostheses according to the sex of the patients (*p* = 0.0161).

Type of Prosthesis		Maxillary	Lover Jaw	Total
Men	Hybrid fixed prosthesis	12 (92.3%)	13 (100%)	25 (35.2%)
	Full fixed rehabilitations	1 (7.7%)	0 (14.4%)	1 (12.9%)
Women	Hybrid fixed prosthesis	7 (50%)	12 (85.7%)	19 (46.3%)
	Full fixed rehabilitations	7 (50%)	2 (14.3%)	9 (5.5%)
Total		27 (50%)	27 (50%)	50 (100%)

Note: *p*-values below 0.05.

**Table 6 ijerph-19-13162-t006:** Marginal bone loss (1.51 ± 1.16).

Variables			*p*-Value
Age	≤65 years	>65 years	0.3466
	1.19 ± 1.39	1.82 ± 0.84	
Sex	Men	Women	0.9797
	1.57 ± 1.20	1.46 ± 1.16	
Tobacco	Smokers	Non-smokers	0.3805
	2.00 ± 1.58	1.43 ± 1.0	
Diseases systemic	+	−	0.0486 *
	2.14 ± 1.02	1.30 ± 1.15	
Clinical follow-up	<36 months	>36 months	0.0820
	1.08 ± 1.14	1.86 ± 1.09	

*: significant *p* value (*p* < 0.05).

**Table 7 ijerph-19-13162-t007:** Distribution of patients (*n* = 10) and implants (*n* = 31) with peri-implantitis.

Variables				*p*-Value
Age		≤65 years	>65 years	
	Patients	5 (38.4%)	5 (35.7%)	0.9180
	Implants	14 (9.3%)	17 (10.9%)	0.8155
Sex		Men	Women	
	Patients	5 (38.4%)	5 (35.7%)	0.9180
	Implants	13 (8.8%)	18 (11.4%)	0.8676
Tobacco		Smokers	Non-smokers	
	Patients	4 (100%)	6 (26.1%)	0.1476
	Implants	8 (19.5%)	23 (8.7%)	0.0827
Diseases systemic		+	−	
	Patients	6 (87.5%)	4 (20%)	0.0098 *
	Implants	13 (17.3%)	18 (7.8%)	0.0015 *
Clinical follow-up		<36 months	>36 months	
	Patients	4 (33.3%)	6 (40%)	0.0534
	Implants	6 (4.4%)	25 (14.7%)	0.3834

*: significant *p* value (*p* < 0.05).

## Data Availability

Not applicable.

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
