# Peer review of "Immediate Functional Loading with Full-Arch Fixed Implant-Retained Rehabilitation in Periodontal Patients: Clinical Study"

_ijerph, 2022, doi:10.3390/ijerph192013162_

Round 1

Reviewer 1 Report

Dear Authors,

Your manuscript is really interesting and well conducted.

Unfortunately it cannot be published in present form and it needs to be revised.

- It is necessary to revise the English used, because in several places the construction of the sentences is confusing.

- The introduction section is very short and is needed to add other references to increase the quality of the manuscript. 

Telescopic overdenture on natural teeth: Prosthetic rehabilitation on OFD syndromic patient and a review on available literature PubMed ID 29460531
Prosthodontic Treatment in Patients with Temporomandibular Disorders and Orofacial Pain and/or Bruxism: A Review of the Literature https://doi.org/ 10.3390/prosthesis4020025 

Oral-facial-digital syndrome (OFD): 31-year follow-up management and monitoring PMID: 29460530

-About the Title of the article, I suggest you to modify it and add the type of article

- I suggest you add a table with the list of abbreviations used in the text.

-Please expand conclusion section with main results and future perspectives of this study

Thank You,

Kind Regards

Author Response

Dear reviewer

Thank you very much for all your comments, attached file in pdf format

José López

Reviewer 2 Report

This clinical study aimed to evaluate the clinical outcomes of immediate loading of implants with totally fixed rehabilitations in periodontal patients. Some considerations will be described below.

Although the authors have mentioned important references, some relevant ones were not included. I suggest that authors update the literature review, including relevant references, such as Guarnieri et al. (2021) doi: 10.11607/prd.4674, Chrcanovic, Albrektsson, Wennerberg (2014) doi: 10.1016/j.jdent.2014.09.013 and Ramanauskaite et al. (2014) doi: 10.1097/id.0000000000000156. 

Line 71.

What criteria were used to define "periodontal patient"?

If  patients required extractions, they could not be initially classified as totally edentulous. 

Line 85

Bone regeneration techniques were performed in the study. The authors were probably referring to techniques prior to implant placement ("no need for bone regeneration techniques"), which should be explained in the text.

Line 100. 

Although there is no consensus in the literature about the need of using antibiotics nor which drug and dosage would bring more benefits in clinical outcomes, I suggest that the authors add a reference to the adopted regimen.

Line 108

Was not reaching a minimum torque of 35N considered an exclusion criterion?

Line 133

The "success criteria" is the same of survival rate (which  is an outcome widely used in follow-up studies)?

Line 156

In all tables, authors must describe to which statistical tests the p-values ​​refer. In addition, the referred indication of variables with p<0.05 is not clear in the tables.

Line 278 

What was the criteria for peri-implantitis case definition? 

Line 309

Is the term "postextraction implants" appropriate for all the sample, since depending on the prosthodontic guided plan, the implants could be inserted in healed bone?

Line 382

The authors dedicated a paragraph to the discussion about the type of prosthesis, whether cemented or screw-retained. However, no comparison was made between them in the statistical analysis, nor was it clear whether the type of prosthesis would influence the outcomes analyzed.

Line 415

I suggest that the authors improve the topic that discusses peri-implantitis, with robust prevalence studies presented in the literature (such as French, Grandin and Ofec, 2019) Thus, one can better discuss whether the results achieved in these periodontal patients are above expectations for individuals without previous periodontitis. Some case definitions of peri-implantitis are also presented in the literature. Authors must explain why they consider what is described in the text.

The authors did not discuss the biological plausibility that would explain a possible worse performance of implants installed in patients with previous periodontitis.

Moreover, they do not discuss the influence of the prosthetic connection used on marginal bone loss, or whether if platform switching was performed. 

In addition, they do not mention in the discussion the limitations of the study. Some patients/implants were evaluated in a short follow-up period of up to 15 months, which could influence the interpretation of outcomes presented. This is a retrospective study that, due to its design, has some limitations. Grouping different types of prosthesis, materials and forms of retention, and installation protocols, in healed sites and fresh sockets, can represent a heterogeneity that negatively impacts the analyses.

Conclusions

If the authors consider that peri-implantitis is more prevalent in periodontal patients, although a 100% survival rate has been achieved, this is an important data for this group of patients and could be mentioned in the conclusion.

Author Response

(The authors gave the same response as above.)

Reviewer 3 Report

1. This study included 305 implants in 27 patients. It is a fair sample size. Considering all periodontally involved patients, it was quite pertinent. The results pointed out 100% survival rate but there were also 31 implants (10.2%) in 10 patients (37%) diagnosed peri-implantitis. No matter it was 10.2% of implants or 37% of subjects, both of them were significant proportion. It should not be concluded a predictable procedure but should be addressed the possible risk of peri-implantitis. Moreover it should be very interesting to know how the author managed those implants with peri-implantitis. This article will be more valuable to discuss the issue and to propose methods for prevention.

2. Criteria used for the diagnoses of peri-implantitis shall be described. For those implants presenting peri-implantitis, the detail of the defects and the tissues could be also valuable to see how the treatment modality may lead to the complications.

3. The results also indicated that 5 patients (18.5%) showed certain prosthetic complications. It is almost 20% of the subjects, considered relevant. Solutions to overcome those complications shall be taken into consideration and discussed. 

Author Response

(The authors gave the same response as above.)

Reviewer 4 Report

Thank you for the opportunity to review the paper titled “ Immediate loading with totally fixed rehabilitation of implants  placed in periodontal patients “

In my opinion the title is very confusing and does not reflect the information in the paper. What are  "periodontal patients" ? Authors must be more specific about what exactly group of patients they want to  examine.

The introduction is very poor and should be much more extensive with up to date references as the information and literature is very old and well known and a lot of papers regarding this topic have been published in recent years.

The nomenclature and description the author used is not a scientific one for example :

Totally-arch- should be full arch

Immediate loading – should be immediate functional loading

Periodontal patients – should be periodontally compromised … or with history of periodontal …

Lines 56-60 – authors should present the subject more precisely as they used very old references

The paper is difficult to read because of a lot of language mistakes, thus it needs extensive correction.

There is no ethical committee approval number

“This study included periodontal patients with partial or total edentulism who required treatment with extractions of all remaining periodontal teeth and immediate dental implants”- this sentence makes no sense

Inclusion/exclusion criteria are completely wrong. For example, inclusion is “no need for bone regeneration techniques.” And the Fig.2 or the data in the text about GBR. Authors forget to put into the inclusion criteria diagnosed periodontal problems. Once they write minimum 30 Ncm next the 35Ncm as minimum IT.

Line 109 – need correction

“. Bone bovine graft Cerabone® (Bottiss Biomaterials GmbH (Zossen, Germany) was applied in all cases.”-  this change the whole study

There is no precise information or description of “Hybrid fixed“ prosthesis (there are many types) , how were they fixed ? With 11 implants per patient the risk of peri-implantitis is very high. Did the authors remove the prosthesis during check ups to control each of them ?

The tables are hard to understand , what is “Jaw” in table 5. – you mean mandible or lower jaw

Why do all patients receive prophylactic antibiotic therapy ??? This makes no sense in edentulous patients.

The discussion needs to be improved .

The inconsistency in the data and diversity of the patients, implants and prosthetics makes the result completely not credible from a scientific point of view.

When submitting papers to high IF journals like IJERPH  we expect major clinical importance and novelty in order to justify the publication. The presented paper has some potential but in this form it is not suitable for publication. With so many authors I would expect a much more expanded and perfectly written paper.

Author Response

(The authors gave the same response as above.)

Round 2

Reviewer 4 Report

Thank you for the opportunity to review this article again. In my opinion, the authors did not manage to significantly improve its quality. They did not follow the comments of the reviewer. Additionally, they have made some changes that are not only confusing but ethically questionable. In the first version, all patients had bone regeneration procedures, in the current version this was changed to "those who needed". Regarding the antibiotics used in all cases, they referred to the article in which the doses are different than they used themselves. In the description of the implant assessment it is stated that the stabilization of the implants was assessed, however, in cemented works it is impossible and in the case of screwed ones it requires removal of the works. The number of understatements and errors completely disqualifies this article for publication. This article is very poor and has minimal scientific value , there are no justification to publish it in a high impact factor journal .

Author Response

Dear reviewer

We thank you for all the time you have spent reviewing our manuscript, allowing us to improve it, and we are enclosing a letter in response to your questions. Thank you very much

José López
